# Experiences of rehabilitation one year after breast cancer diagnosis–A focus group study from the ReScreen randomized controlled trial

Ing-Marie Olsson[1,2]*, Charlotta Dykes[1], Lisa Rydén[2,3], Ulrika Olsson-Möller[1,4], Marlene Malmström[1,2,5]

1 Department of Health Sciences, Lund University, Lund, Sweden, 2 Skåne University hospital, Malmö, Sweden, 3 Department of Clinical Science Lund, Surgery, Lund university, Lund, Sweden, 4 Department of Nursing and Integrated Health Sciences, Kristianstad University, Kristianstad, Sweden, 5 The Institute for Palliative Care, Lund University and Region Skåne, Sweden

* Maivor_ing-marie.olsson@med.lu.se

**Data Availability Statement:** The data for this study consists of interview transcripts that contain sensitive personal information. Due to restrictions

## Abstract

### Background

Treatment for breast cancer poses major challenges and leads to a variety of side-effects and problems that affect life for a long time. Experiences and symptoms vary, and research indicates a lack of structures for ensuring individualized rehabilitation. This qualitative focus group study aims to explore the experience of women with BC after participating in a complex randomized controlled trial (RCT) (Clinicaltrials.gov NCT03434717) focusing on cancer rehabilitation from a comprehensive perspective.

### Method

Nine semi-structured focus group interviews with women (n = 30) who participated in the Rescreen RCT were conducted. The women were interviewed divided into three RCT groups (intervention, control, or observation group). Data were initially analyzed inductively using conventional content analysis, followed by a deductive approach, guided by the result from the inductive analysis.

### Results

The inductive analysis resulted in two categories and four sub-categories and showed a great variation in experiences and needs. Some women described a well-functioning process, while others described lack of individualized information, continuity with healthcare providers, and clear pathways for support. After the deductive analysis, a variation between the groups appeared. Women in the intervention group expressed that a proactive and individualized approach facilitated rehabilitation and they experienced a feeling of being recognized as a person, which they highlighted as important. On the contrary, women from the control group described feelings of being abandoned from healthcare, hindering

imposed by the Swedish Ethical Review Authority and the consent given by participants, we are unable to share these data publicly. Participants agreed to participate exclusively in the ReScreen study. However, de-identified excerpts are included in the paper. Data sharing is available on reasonable request. Requests for additional information can be directed to the Swedish Ethical Review Authority at registrator@etikprovning.se.

**Funding:** "The ReScreen study (MM) was funded in all by external funding through governmental funding of clinical research within the Swedish National Health Service (ALF funding) (award number: 200-Project0190), FORTE (https://forte.se/en/) (award number: 2020-00105), the Swedish Breast Cancer Association (https://brostcancerforbundet.se), the Cancer and Allergy Foundation (https://cancerochallergifonden.se) (award number: 220) and The Swedish Cancer Society (https://www.cancerfonden.se/om-oss/about) (award number: 230656FE). The funding bodies had no role in any components, from study design, to submission for publication. There was no additional external funding received for this study. For the purpose of Open Access, the author have applied a CC BY public copyright license to any Author Accepted Manuscript (AAM) version arising from this submission".

**Competing interests:** The authors have declared that no competing interests exist.

**Abbreviations:** HCS, Healthcare System; BC, Breast Cancer; QoL, Quality of Life; MRC, Medical Research Council; RCT, Randomized Controlled Trial; IG, Intervention Group; CG, Control Group; OG, Observational Group; IN, Intervention Nurse; HCP, Healthcare Providers; PCC, Person-centered Care.

rehabilitation. The observation group expressed that their needs had been fulfilled within the healthcare system.

## Conclusions

This study adds important knowledge to the evaluation of the ReScreen model and contributes to existing research on how individualized rehabilitation after breast cancer can be applied in clinical practice. A proactive, person-centered approach in rehabilitation, aimed at those with extended needs, would potentially optimize rehabilitation and facilitate the recovery process after breast cancer treatment.

## Background

Despite growing evidence of the importance of cancer rehabilitation and established national cancer rehabilitation guidelines, cancer rehabilitation plays a marginal role in today's medicine- and treatment-driven healthcare system (HCS) [1, 2]. Today, breast cancer (BC) is the most common cancer for women worldwide [3], with 8600 women diagnosed in Sweden annually [4]. During the last decades, improved prognosis has satisfyingly resulted in a growing number of survivors facing life after BC [5, 6]. Even though most women recover and regain levels of quality of life (QoL) comparable to the general population, some women experience long-term persistent and burdensome side-effects related to diagnosis and treatment [7]. Despite extensive research, women with BC still report that they receive no or inadequate rehabilitation [8], potentially hampering their recovery. Rehabilitation remains crucial, motivating the need for increased knowledge of BC and individualized cancer rehabilitation and how this can be ensured in clinical practice.

The definition of cancer rehabilitation clearly emphasizes the importance of ensuring rehabilitation from a comprehensive perspective and incorporating a focus on the patients' physical, psychological, social, as well as existential needs [9]. Cancer rehabilitation aims to maintain or regain the ability of functioning, activity, and QoL despite consequences of the cancer disease and related treatment. Despite this broad definition, earlier interventions have predominately been focusing on one of these aspects [10], which suggests that there is still a considerable lack of research that focuses on cancer rehabilitation from comprehensive perspectives.

In research focusing on the experiences of BC, the women describe a striving to obtain normal life [11, 12]. During this striving, a multitude of problems and symptoms may occur, including pain, fatigue and concentration impairments even years after diagnosis [13], as well as sleep, weight, and menopausal/sexual problems [7]. In addition, challenges such as distress [14], depression [15], and fear of cancer recurrence [16] are frequently described. Evidence shows that consequences from BC and related treatment affect QoL [13, 17, 18], and that physical changes affect identity [19, 20]. Body image concerns are prevalent in women with BC, limiting daily life and social functioning [20]. Body image encompasses several behavioral issues, including checking behaviors for signs of cancer, impaired social and intimate relationships including sexuality, which negatively affect women´s emotional wellbeing [21]. Lower functioning in daily life compared to the general population are reported [13], and role disruptions and struggles in resuming meaningful activities are described [22]. Additionally, factors such as comorbidity [23], social vulnerability [24], and age [25] have shown to affect the

individual need for support, clarifying the need for individualized comprehensive rehabilitation to recover.

Previous research suggests that important factors for recovery include: support from family and one's social network [19, 26–28], continuity with healthcare providers (HCP) [29, 30], support to navigate the HCS [30, 31], as well as timely and individualized information during the cancer trajectory [27]. Support including a proactive approach to rehabilitation [32, 33], and information concerning the importance of and access to rehabilitation [32, 34] seem to facilitate recovery. Interventions focusing on psychological aspects of cancer rehabilitation have shown to decrease the level of anxiety and increase QoL [35], and to reduce body image issues [36]. Some of the psychological interventions that show promising results for stress reduction and wellbeing include mindfulness-based stress reduction interventions [37] and self-compassion interventions [38]. Large scale studies focusing on physical activity interventions showed a long-term effect on the burden of fatigue [39], emotional symptoms [40], and increased health-related QoL during and after oncology treatment [41]. Several interventions have been shown to affect more than one symptom [10] and adding the individuality to the experience further illuminates the importance of individualized supportive initiatives throughout the cancer process [11, 20, 42, 43]. Despite all this knowledge, absence of systematic assessment for needs among patients with cancer has been demonstrated [44], and studies have indicated lack of structures as potentially affecting BC patients' access to rehabilitation [34, 45, 46].

Identifying women with extended rehabilitation needs early in the cancer process is essential for optimal recovery. Recent studies have shown that a self-reported level of distress may be a valid indicator of extended rehabilitation needs [14]. The definition of distress in the field of cancer describes a multifactorial and unpleasant experience of physical, social, spiritual and/or psychological nature that may interfere with the ability to handle cancer, physical symptoms, and treatment [47]. Distress has been shown to predict poorer social adjustment [48] and non-adherence to cancer treatment, potentially affecting cancer recurrence and long-term survival [49]. Therefore, the level of distress needs further investigation to evaluate its clinical applicability. Potentially, this would be an important factor to enable evidence-based individualized interventions to be introduced aimed at those who need it the most.

It is recognized that cancer rehabilitation needs are complex and vary greatly between individuals and over time. Therefore, complex interventions, which often include several active ingredients, are needed to ensure individualized rehabilitation. The Medical Research Council's (MRC) framework for complex interventions emphasizes that, to understand the ingredients of a complex intervention, quantitative as well as qualitative research contribute to the understanding of results [50]. Focusing on participants' perceptions and experiences can give a comprehensive understanding of the active ingredients of the intervention. This qualitative focus group study aims to explore the experiences of women with BC after participating in a complex randomized controlled trial (RCT), focusing on cancer rehabilitation from a comprehensive perspective.

## Method

### Design

This explorative focus group study is a part of the ReScreen complex intervention study [51] (described in brief below). The focus group study is reported according to the Consolidated Criteria for Reporting Qualitative Research (COREQ) guidelines [52] (S1 File).

## The ReScreen study

The ReScreen study (Clinicaltrials.gov NCT03434717) was developed according to the MRC framework for complex interventions [53] with the overall aim to develop and, in a three-armed RCT, test and evaluate a model for comprehensive screening-based individualized rehabilitation following treatment for primary BC [51]. The MRC framework incorporates four phases: 1) development, 2) pilot and feasibility testing, 3) evaluation and 4) implementation. This focus group study is a part of the third phase focusing on the evaluation of the RCT from a qualitative perspective.

Based on previous research [14, 54] and the ReScreen pilot and feasibility study [55], the participants' level of distress was assessed using the validated Distress Thermometer (ranging 0–10) with a cutoff of $\geq 5$ as an indicator for extended rehabilitation needs [47, 55, 56]. Participants with high distress were randomized into the intervention group (IG) or control group (CG), while participants with low distress formed an observational group (OG). The CG and OG received care as usual, including follow-up by a surgeon/oncologist, contact with a nurse, and a physiotherapist or social worker when appropriate. In addition, the IG was monitored proactively by an intervention nurse (IN) with a specific assignment to, during the one-year follow-up period, facilitate comprehensive evidence-based individualized rehabilitation supported by a decision support tool developed in the first phase of the ReScreen study. The decision support tool was used when structuring an individual rehabilitation plan, in agreement between the patient and the IN, focusing on the participants' health-related behaviors and comprising clinical and evidence-based knowledge concerning known problems related to BC treatment such as fatigue and anxiety. The decision support tool consists of four steps: 1) general advice about rehabilitation and exercise; 2) structured needs assessment; 3) matching evidence-based interventions to individual needs; 4) developing a rehabilitation plan, setting goals, and planning follow-up. The IN provided support and advice inspired by motivational interviewing, and supported navigation through coordination of contacts and referrals due to the individual rehabilitation needs. Evaluation of the rehabilitation plan and follow-up was based on the patients' individual desires and needs, with a minimum of contact at two weeks after the start of treatment, and once a month the first three months, and then based on individual needs up to one year after diagnosis. The intervention, as well as the decision support tool, are described in detail in a published protocol [51].

For clarification, throughout this manuscript abbreviations for the three groups are used as follows: OG for observational group (low distress before the start of treatment), CG for control group (high distress before the start of treatment, NOT receiving the intervention), and IG for intervention group (high distress before the start of treatment, receiving the intervention).

## Context

The study was conducted at one university hospital and one county hospital in the south of Sweden. According to the Swedish "National cancer rehabilitation guidelines" [9], cancer rehabilitation is organized in three levels: basic, specialized, and advanced. On the basic level, where the ReScreen study was conducted, "care as usual" is represented by nurses that are the patients' primary contact at the out-patient unit during the pre- and post-treatment phases. Their responsibility is to provide information, assess needs, and initiate rehabilitation actions. There are structures for rehabilitative follow-up, based on medical treatment, such as appointments to a physiotherapist after axillar lymph node dissection or to social workers when in need of psychosocial support. Rehabilitation needs not related to the medical treatment regime are identified depending on the patients' or nurses' initiatives with referral to specialized rehabilitation. If complex needs are identified, the patient can be referred to an advanced cancer

rehabilitation unit that includes a multi-professional team that focuses on cancer rehabilitation [10]. Structures for identifying needs and referral are, despite national cancer rehabilitation guidelines [9], inconsistent and sometimes lacking in clinical practice [45].

## Recruitment and participants

Participants who had fulfilled their one-year participation in the ReScreen RCT were eligible for the focus group study. Already at inclusion in the ReScreen RCT, they were informed that they could be invited to a group interview regarding their experiences of rehabilitation. The inclusion criteria in the ReScreen study were women diagnosed with primary BC planned for surgery, age ≥18, and ability to communicate in Swedish. Exclusion criteria were recurrent disease, pregnancy, and inability to participate due to cognitive impairment, severe mental illness, or drug addiction. Between 12 April and 14 November 2022, participants from the ReScreen RCT were purposefully invited to the focus group study intending a variation regarding participation in IG, CG or OG. The inclusion criteria for the focus group study required completed participation in the ReScreen study and initiation of treatment within the past two years prior to inclusion. The first author (IMO, female) approached the participants by phone and asked if they wanted to participate in a focus group interview. Out of 109 eligible participants, 95 could be reached and 42 accepted participation. After cancellations due to individual reasons a total of 30 women participated. For participants' characteristics, see Table 1.

All participants signed written informed consent before the interview. The study was approved by the Regional Ethical Review Board in Lund, Sweden (reference number 2015/505, amendments 2018/924, 2020–04664).

## Focus group interviews

Nine focus group interviews with 2–6 participants in each group were conducted during 2022. Variation regarding participation in IG, CG or OG during the ReScreen study was achieved, with 12 participants being from IG (three focus groups), 8 from CG (three focus groups), and 10 from OG (three focus groups). The interviews were carried out in a separate room at the hospital and lasted between 70 and 130 minutes. A semi-structured interview guide focusing on the patients´ experience of rehabilitation during the first postoperative year was used (Table 2). The interviews started with one of the interviewers explaining the definition of cancer rehabilitation, followed by the open-ended question "What are your thoughts of rehabilitation during the year after the diagnosis?" Two researchers, one PhD student (IMO or CD,

**Table 1. Participants' characteristics.**

| | |
|---|---|
| **Age, m (range)** | 59,7 (32–84) |
| **Months since start of treatment, m (range)[1]** | 16,4 (12–24) |
| **Working, n** | 20 |
| • ≥ 75–100%, n | 17 |
| • < 75%, n | 3 |
| **Not working, NO, n** | 10 |
| **Living situation[1]** | |
| • Married/Cohabitant, n | 18 |
| • Living apart, n | 1 |
| • Living alone, n | 10 |
| **Living with child < 18 years, n** | 8 |

[1]one missing

**Table 2. Semi-structured interview guide.**

| Opening question | What are your thoughts about rehabilitation during the year after the diagnosis? |
|---|---|
| The experience of receiving rehabilitation from healthcare | • What type of support did you need to manage the situation?<br>• What type of support were you offered?<br>• What type of support did you seek?<br>• How did you get in touch with healthcare regarding rehabilitation?<br>To what extent has the rehabilitation offered met your expectations? *This includes care, adequate information, and attention to physical and mental well-being.*<br>• What information did you receive concerning rehabilitation, and what information did you seek for yourself? |
| The experience of structure and collaboration within the healthcare system | • How did healthcare cooperate regarding your rehabilitation?<br>• What goals were set, and were these goals followed and evaluated?<br>• To what extent do you feel that you had the opportunity to participate in decisions about your rehabilitation? |
| Important persons and availability of healthcare | • Who has been the most important person during this journey?<br>• Who has supported your recovery?<br>• In what way was healthcare available or unavailable?<br>• What or who would you have needed in addition to what was offered? |
| Final question | What factors have contributed to your sense of meaning and hope throughout this journey? |

female) and one senior researcher (MM, female or UOM, female) with predefined roles were present at each interview. One interviewer was moderating the interviews and supported the participants to focus on the aim, while the assistant interviewer kept notes and asked probing questions. The assistant interviewer also summarized the interviews in the end of each interview to allow immediate member checking [57]. The information that was revealed was discussed between the two interviewers directly after the interviews, followed by notes describing the experience according to the interview situation and feelings. After the ninth interview, it was decided that the data collection would be closed as no new information emerged.

## Data analysis

The interviews were recorded digitally and transcribed verbatim. Before the start of analysis, the first author (IMO) validated the transcripts by reading the text and listening to the interviews simultaneously. Thereafter, the interviews were analyzed using conventional content analysis with an inductive approach, allowing categories to flow from the data [58]. The transcribed interviews were repeatedly read by four authors (IMO, CD, UOM, MM) to get a sense of the whole, followed by a joint discussion of the interviews. The first two authors (IMO and CD) had the main responsibility for the analysis following the procedures for conventional content analysis as described by Graneheim and Lundman (2004) [59], while MM and UOM ensured the link between the data and the analysis. The authors had varying experiences of qualitative research ranging from limited to extended experience. The text from all interviews were initially analyzed as a whole and all the text was coded [60] through abstraction of condensed meaning units by the first two authors separately. The codes that answered the aim of the study were then sorted into sub-categories and categories, representing the manifest meaning [59], with a back-and-forth process between the interview text, codes, subcategories, and categories to validate the interpretation. Recurring meetings were held in the process to ensure agreement in coding, subcategories, and categories. Two categories and four subcategories

were identified. A breadth and variety in experiences of the rehabilitation phase were illuminated. To deepen and increase the knowledge of the variation within the three groups separately (IG/OG/CG), the analysis proceeded by adopting a deductive approach in which the categories and subcategories were specifically illuminated in each group. Finally, agreement between all of the authors on the latent meaning in the interviews was formulated as an overall theme [59].

## Result

### Theme: Being an individual in a complex healthcare system

The analysis resulted in two categories and four sub-categories captured in the overall theme "Being an individual in a complex healthcare system". This theme illuminates how the women found themselves in a complex and sometimes fragmented HCS. In this context, the medical process, in terms of examinations and treatments, was often experienced as predetermined and well-structured, while support for rehabilitation was found to be unstructured and deprioritized. In this unfamiliar and stressful situation, the importance of being seen as an individual and receiving individualized and time-adapted support was stressed as substantial. Women's experiences varied greatly, both on an individual level and between the three groups (IG, CG, OG). This was illustrated in experiences of encountering both a well-functioning HCS that met their support needs and experiences of not receiving the support they needed, leading to feelings of uncertainty in relation to recovery. The variation is captured in the two main categories "Feeling safe—a prerequisite for rehabilitation" and "The individual needs as guidance for rehabilitation" (Fig 1).

| THEME | | | |
|---|---|---|---|
| Being an individual in a complex healthcare system | | | |
| CATEGORY | | CATEGORY | |
| Feeling safe - a prerequisite for rehabilitation | | The individual needs as guidance for rehabilitation | |
| SUB-CATEGORIES | | SUB-CATEGORIES | |
| To be able to navigate and understand the healthcare system | To be acknowledged and valued in the healthcare system | To be able to adjust to and recover in daily life | To receive support when needed |

**Fig 1. Overview of theme, categories, and sub-categories.**

## Category 1: Feeling safe—a prerequisite for rehabilitation

The new situation that the women faced following a BC diagnosis was experienced as challenging. They expressed the importance of gaining understanding of the system, receiving individualized information, and experiencing trust and continuity in their recovery. These components collectively played a crucial role in their satisfaction with the rehabilitation process, as they made the women feel secure and regain control of the situation.

**Sub-category 1: To be able to navigate and understand the healthcare system.** Understanding and navigating the HCS, in terms of having knowledge about the organization and available support, was perceived as crucial for feeling secure during the rehabilitation process. Needs associated with recovery and rehabilitation were described as varying throughout the cancer process. This meant that the women often experienced different challenges and needs over time, and that the burden of these fluctuated. The ability to handle these challenges was often described as linked to knowing where to seek support when problems or questions arose. As a consequence of this, thoughts regarding the roles and responsibilities of different HCPs were expressed, with questions arising regarding who was responsible for providing assistance and when, whether it was the responsibility of the patient or the HCP. Another aspect, experienced particularly by women with high distress, concerned situations where gaps in the structure between clinics became apparent. The women repeatedly cited instances when communication methods and pathways were unclear, posing a threat to their short- and long-term rehabilitation, as they were unsure where to seek help. When the women had limited knowledge of how the cancer pathway was organized and how they could navigate it, this contributed to a sense of alienation, leading to uncertainty and increased need for support to navigate efficiently.

The women concurred that the period after completing active treatment was particularly challenging, and they often expressed a desire for support beyond current one-year follow-up structures. When expectations for support with navigation appeared as unmet, the women in the CG expressed feelings of disappointment and uncertainty. One woman described her experience:

> *"What is also strange, and it is also part of the rehabilitation I think, is that you start here, or at least I did, in the breast cancer unit. Then you are discharged, but you don't quite understand that. I called at some point to my contact nurse, "No, you are no longer under our care" Oh "I´m not?". Oh." (CG:3)*

Expressions of insecurity also emerged regarding access to adequate and accurate information from HCP. Women experiencing high distress often found the information to be impersonal and inadequate on a personal level, lacking the specificity needed to address their rehabilitation needs. When information provided early in the process was altered or proven incorrect, such as regarding planned treatment or details about preparations before treatment, a feeling of distrust arose. As a result, women did not always seek help when needed, hindering their rehabilitation process. In contrast to the women in the CG, those in the IG had experienced a sense of security through the communication, information, navigation, availability, and trust provided by the IN. One woman expressed it like: *"And so you decided a new day and a new appointment, how often I wanted . . .(. . .). . . I thought that was great.." (IG: 2)*. Similarly, other women described how the proactive approach by the IN provided her with the information she needed, highlighting the importance of individualized support:

> *"Without that contact with (IN), I would probably have been quite lost. Then I probably would have had a lot of questions, and maybe I wouldn´t have been able to get that. . . yes,*

*that contact and that sense of security, actually, to ask her if I haven't been feeling well."*
*(IG: 3)*

The women in the OG, identified as having low distress at diagnosis, expressed overall well-functioning communication with HCP regarding information and navigation, and their individual needs were often met. This facilitated their prerequisites for rehabilitation and generated a sense of satisfaction, often leading to no further requests for extended support.

**Sub-category 2: To be acknowledged and valued in the healthcare system.**   The need to feel recognized, trusted, and to have continuity in the relationship with HCPs was expressed as essential for fostering a sense of inclusion and personal significance. The women expressed a need to be known and to know the HCP who they expected to be aware of their individual needs while guiding them through their rehabilitation process. The women with high distress described the initial encounter with the nurse as crucial and as the starting point for their rehabilitation process. This relationship, which they wished to maintain throughout the rehabilitation process, was felt to be important for feeling recognized, important, valued, listened to, and secure. It also helped them in handling unforeseen problems along their rehabilitation process. Lacking this continuity, one woman from the CG described:

*". . .there was no continuity, so there were different doctors, different nurses, you know. And they had no idea who I was as a person. No, that's how I feel. . ."()" I think the feeling it might give, or gives. . . is that you´re not. . . you know, you´re not really included." (CG:1)*

Through the intervention, the women in the IG expressed that having access to the same dedicated nurse continuously contributed to them feeling recognized and confirmed as individuals, and they didn't have to constantly repeat information about their well-being. They described how they felt that the IN got to know them and their situation, a key factor to feeling secure:

*"And then when (IN) called every month, she would ask, "So how did that visit or conversation go? So you never really had to explain again, she just knew. (.... .) It feels like she knows you. She knows what has happened, and she has probably gone in and read the records as well, so she is prepared to ask questions based on the situation you´ve been through". (IG:3)*

Encountering kindness and understanding from HCP was also repeatedly described by the women with low distress (OG). However, they experienced that their rehabilitation needs were often met on a personal level and emphasized that continuity was positive, although not yet as crucial for their recovery compared to the women with high distress.

## Category 2: The individual needs as guidance for rehabilitation

The individual rehabilitation needs and the ability to cope with the new situation affected daily life on several levels. The women strove to feel capable of handling the situation and emphasized the importance of knowing what to expect during the rehabilitation process. When support from healthcare was lacking, feelings of being left alone were described, and it was evident that there was often an unmet need for proactive and timely rehabilitation support. For the support to be perceived as adequate, it was necessary for it to be individualized and timed according to women's needs.

**Sub-category 1: To be able to adjust to and recover in daily life.**   Receiving individualized support and engaging in everyday activities, such as returning to work and participating in leisure activities, were described as motivating and important when struggling to adjust to

the new life situation. However, when experiencing side-effects from medical treatments such as hot flashes, hair loss, fatigue, and pain, it influenced their identity and ability to recover. Therefore, the women expressed the importance of knowing what to expect as normal side-effects and reactions, to be prepared and able to handle upcoming problems in daily life. The sometimes paralyzing fear of cancer recurrence was also mentioned repeatedly, and thoughts about the future were integrated into daily life after BC, hindering mental recovery. Throughout the rehabilitation process, the safety net provided by family and friends was described as crucial, while the ability to ask for help and the need for support from HCP varied. Women in the CG described that the feeling of being responsible for their own rehabilitation in daily life resulted in a feeling of being isolated and unable to cope with the situation. One woman from the CG expressed how the lack of the support she needed affected her rehabilitation abilities:

> ” Because if you are constantly going to fight and run into obstacles, then you can't take it, you ignore it, and then you lie here and feel like shit” (CG:3)

The women expressed that one's own attitude towards the situation, as well as previous positive experiences with illness and healthcare, could have positively affected how they adjusted to the new life situation and the likelihood of seeking support. In contrast, women also described how earlier negative experiences could lead to avoidance of healthcare and thus missed opportunities to receive support, such as earlier experiences of not receiving adequate help from healthcare for a family member. While women in the CG often felt left to themselves, those in the IG described how, through the support from the IN, they were able to cope in their daily life. With a proactive approach, the IN helped them to securely start and maintain rehabilitative activities. The IN served as an important resource, through supporting and encouraging the women in various areas of life. The individuality in the rehabilitation process was important, and advice on how to adapt during treatment periods were exemplified, like to remove obligations, or to just go outside to get some air and enjoy nature. Advice on how to integrate rehabilitation in their usual activities was described as helpful to be able to return to their everyday life. For some women this was a long process, and the INs advice were described as a gentle reminder of small steps they had achieved, while others described activities possible to resume early after the start of treatment, illuminating how advice was customized:

> "Yes, I worked out. I had surgery on Thursday and did spinning on Saturday morning. Because they said . . . [IN] said like this, . . . Now, you go out [laughter]. Get into life, start working, spend time with people and go spinning and do what you. . . so." (IG:2)

When attempting to return to daily life, attitudes towards life and earlier positive healthcare experiences influenced the recovery of women with low distress (OG), resulting in a reduced need for additional support and rehabilitation beyond usual care.

**Sub-category 2: To receive support when needed.** Throughout the individual process, from diagnosis, the experience of treatment, side-effects, as well as dealing with situations in one's private life and at work, the timing of rehabilitative support was crucial, but not always accommodated. When available resources and structures for rehabilitation did not align with the women's individual rehabilitation needs, the risk of not receiving support in time increased. In contrast to those who did not articulate their rehabilitation needs, those who did so gained access to extended support from advanced cancer rehabilitation services more often. The women in the IG specifically expressed that the proactive approach of the IN, which included early identification of needs, relevant information about rehabilitation, and referral to advanced cancer rehabilitation resources when needed, was fundamental to meet their

individual rehabilitation needs. This was illuminated in a conversation between the women in one of the intervention groups:

> Women 1: "And then this to verify and ask "how is it going" . . . when she called. It was really good to have that. Woman 2: "Yes, I thought that was good too". Women 1: "Yes, to know that now she will call, and then I can be sure to ask this or. . ." Woman 2: "And then you always had thought things through, so you had written down some things to ask". Woman 3: She also asked "shall I call in four weeks or what do you want?" that I thought was very good"-. . .(. . .). . . Woman 4: "I had more like existential, because I needed that to, someone to talk to . . .(. . .). . .was worried about how it affected my next of kin, because I couldn't talk to them about that". (IG:1)

The women in the OG and CG described how they handled upcoming challenges by themselves but with one vital difference. Unlike the OG, the women in the CG felt that they were left alone in their doubts, as they were expected to personally manage both urgent physical and psychological problems. This led to unidentified and unmet rehabilitation needs, especially during periods without planned visits to the healthcare. One woman suggested a way in which her needs could have been identified in time:

> "I believe that in my case . . . I sometimes need time to think and let things sink in. So if someone had caught up with me maybe a week, two, or three weeks later just to give feedback, "How has it been, has anything happened, how do you feel now?" Because there is so much you take in. And you have to process and understand and okay, next step, plan further, what's next". (CG:2)

On the contrary, the women in the OG mainly described how they had received the support they needed for rehabilitation during their cancer journey. However, they also provided examples of how healthcare offered them rehabilitative support, but not always continuously and in time with their needs.

## Discussion

This study shows that the current rehabilitation structures following BC treatment often do not align with women's rehabilitation needs. The overall theme "Being an individual in a complex healthcare system" captured their view of the HCS as complex and often unindividualized. The novelty of this study lays in the inductive and deductive design in which participants in separate RCT groups were asked to share their experiences about rehabilitation. This design allowed multiple perspective to be derived from the data and clearly showed how individual needs were experienced in various ways, depending on their previous experiences and the support they had received during the rehabilitation process. While today´s HCS is organized based on expected side-effects from medical treatment [45], the variation between the groups in this study demonstrates the importance of identifying those in need of extended support based on individual needs rather than the medical aspects. This study indicates that the Distress Thermometer [47, 56] has the potential to be a useful tool in this process, supporting systematic screening for routine use. This study clearly points towards the importance of a comprehensive approach for rehabilitation and how this can be applied in clinical practice.

The study contributes to the existing literature by demonstrating factors that can facilitate the patients' rehabilitation process, consistent with earlier research. These factors include navigation of HCS [31], continuity of care [29], and provision of individualized information [27]. This study also adds several perspectives on the significance of being recognized as a person,

the ability to resume daily life activities, and receiving timed, proactive, and continuous support. These aspects were all described as facilitative and important for feeling secure, while the most crucial aspect was the care being guided by their individual needs, aligning with person-centered care (PCC). PCC entails a focus on the individual process, emphasizing the significance of comprehending what the patient values in their life and how they interpret their experiences [61]. While earlier research emphasizes the importance of individualization and PCC [10, 30], insufficient attention has been given to how this can be effectively ensured and how patients with extended rehabilitation needs can be identified. In this study, the women with low distress (OG) more often described trust in and satisfaction with the HCS than the women with high distress. An important finding was that numerous women with BC appeared to manage their rehabilitation needs within their own ability, mainly their ability to ask for support, and within their own social context. These women had low distress before the start of treatment. On the contrary, it was clearly demonstrated that, for the women with high distress, the need for support required initiative from healthcare, and this supports the introduction of systematic screening for extended needs as a routine praxis early in the trajectory. The core components of the intervention with timely identification of rehabilitation needs, evidence-based interventions, and pro-active follow-up were described as crucial. The women in the IG described how these components had facilitated the process, while the absence of such support was described by the CG as hindering rehabilitation. These findings are interesting but require further investigation and verification through longitudinal studies.

The importance of being able to navigate the HCS was repeatedly discussed by the women in this study, along with the gaps in the healthcare structure that hindered the possibility to receive rehabilitation support. Research stresses the significance of managing to navigate the HCS to cope with the new situation [31], during and after periods of active treatment. A recent study demonstrated how patients with BC felt an overwhelming sense of responsibility to handle everything on their own after primary treatment [62] and our study revealed that organizational structures hindered the women from assuming this responsibility. For instance, when late side-effects appeared, they described a feeling of abandonment when they did not know where to turn. This illuminates shortcomings and gaps in today's follow-up routines that potentially impact women's rehabilitation, since navigation support appears to be an ongoing requirement to ensure optimized rehabilitation, aligning with earlier research [22]. The IG described the proactive approach within the intervention as a key factor in bridging gaps in the HCS and facilitated timely identification of needs. In contrast, the traditional contact structure, reliant on women initiating contact and expressing their own needs, was perceived as a barrier for rehabilitation in patients with higher distress and allocated to the control group. For example, fatigue was described as hindering the ability to initiate contact, leading to missed opportunities for rehabilitation. These findings suggest that the proactive intervention approach was a crucial factor, consistent with recent findings [62], which, due to insufficient structures for follow-up, recommend continuous support and checkups initiated by HCP.

Aligned with the principles of PCC, the women in this study expressed a desire for HCP to possess knowledge about their unique situation, to support their return to daily life, which was crucial for recovery. The needs seemed to be influenced by a wide range of factors, including their unique history, family situation, and individual capacity to manage side-effects and understand the HCS. These individual prerequisites are described in the literature, with the rehabilitation process noted to vary based on the patient's ability to balance and prioritize their everyday life while recovering from BC [43]. In our study, the IG emphasized how continuity with the IN enabled them to feel known and understood within their own ability and situation, while the CG expressed feelings of anonymity. Previous research has described the importance of continuity [29] and how maintaining continuity with HCPs allows patients with cancer to

feel acknowledged as persons and enables them to share their personal stories with HCPs [63]. This finding aligns with our study, where the IG expressed high levels of satisfaction with the continuity they received, which facilitated the development of a trusting relationship. These findings highlight how incorporating the woman's unique situation and everyday life into the rehabilitation plan created a sense of confirmation and recognition as a person, which are core tenets of PCC and crucial for optimizing rehabilitation outcomes in our study. Within a complex HCS that focuses on medical treatment, a comprehensive approach for rehabilitation founded in individual needs and allowing the individual situation to guide timing for support is essential. The focus of the interviews was on experiences of rehabilitation. Treatment consequences and side-effects, such as impact on body image, are not emphasized in this study but are important to address in interventions. A comprehensive, individualized rehabilitation approach has the potential to identify and address these important issues.

## Strengths and limitations

This study has strengths and limitations. The diverse experiences among the three groups are interesting and important findings, suggesting that using level of distress as an indicator for extended rehabilitation needs could potentially be clinically relevant and that the individualized rehabilitation intervention in the ReScreen study might enhance rehabilitation. Qualitative analysis is not traditionally used to compare groups, although it has to some extent been described in the literature [64]. In this study we focused on the variation between groups, aiming to enhance the understanding of the ReScreen intervention. To evaluate the effect of the intervention and the accuracy of using distress as an indicator for extended rehabilitation needs, a comparative longitudinal study that evaluates the effect on distress, quality of life, and associated factors is needed.

The combination of inductive and deductive design strengthened the robustness and the credibility of this study. Credibility was further ensured through meticulous interview techniques and analysis methods, including instant member checking and frequent meetings with co-authors for discussion, to ensure adequate interpretation of experiences.

The study has some limitations. Thirty out of 96 invited participants took part in the interviews, which may affect the transferability and dependability of the findings. Reasons for non-participation were not systematically collected during recruitment due to ethical considerations. Previous research has identified various reasons for non-participation, including practical issues and health status at the time of inclusion [65] which are likely to have also affected participants in our study. The non-participation of these women may have resulted in valuable insights of rehabilitation needs being missed, limiting the applicability of the study in other contexts. Last-minute cancellations led to focus groups with only two participants, which might be viewed as restrictive. However, excluding these participants would have raised ethical concerns, as they still contributed valuable insights that contributed to the rich description in experiences. Variations in focus group size can impact the richness of data, with smaller groups often recommended for topics where participants have intense or lengthy experiences [66]. Despite these limitations, the study's methodological rigor and the depth of nuanced insights gained from qualitative interviews, delving beyond intervention effectiveness [45], contribute valuable knowledge to and understanding of the ReScreen project's evaluation process.

## Conclusion

This study adds important knowledge to the evaluation of the ReScreen model and contributes to existing research on how individualized rehabilitation after breast cancer can be applied in

clinical practice and align with person-centered care. Rehabilitation during and after BC treatment should be based on the individual´s needs and should be evaluated by validated screening tools. The personal process doesn't seem to fully fit into the process offered by the healthcare system which motivates a comprehensive, proactive, and person-centered approach for rehabilitation aimed at those with extended needs. This would potentially optimize rehabilitation and facilitate the recovery process after breast cancer. The variation between groups needs further investigation but serves as an important finding that can contribute to future cancer rehabilitation research.

## Supporting information

**S1 File. COREQ checklist.**
(PDF)

## Acknowledgments

The authors would like to thank all the women who generously shared their stories and contributed to the study with their unique experiences. The authors would also like to thank Helena Erixon, the intervention nurse, who delivered the intervention to the IG with professionalism, a person-centered focus and such a warm heart. Finally, the authors would like to thank both Helena Erixon and Eva Wahlström for all their efforts and hard work with administration of the ReScreen study, and Katarina Velickovic for revision of language of the manuscript.

## Author Contributions

**Conceptualization:** Lisa Rydén, Ulrika Olsson-Möller, Marlene Malmström.

**Data curation:** Marlene Malmström.

**Formal analysis:** Ing-Marie Olsson, Charlotta Dykes, Lisa Rydén, Ulrika Olsson-Möller, Marlene Malmström.

**Funding acquisition:** Marlene Malmström.

**Investigation:** Ing-Marie Olsson, Charlotta Dykes, Ulrika Olsson-Möller, Marlene Malmström.

**Methodology:** Ing-Marie Olsson, Charlotta Dykes, Lisa Rydén, Ulrika Olsson-Möller, Marlene Malmström.

**Project administration:** Ing-Marie Olsson, Ulrika Olsson-Möller, Marlene Malmström.

**Resources:** Ulrika Olsson-Möller, Marlene Malmström.

**Supervision:** Lisa Rydén, Ulrika Olsson-Möller, Marlene Malmström.

**Validation:** Ing-Marie Olsson, Charlotta Dykes, Lisa Rydén, Ulrika Olsson-Möller, Marlene Malmström.

**Visualization:** Ing-Marie Olsson, Charlotta Dykes, Ulrika Olsson-Möller, Marlene Malmström.

**Writing – original draft:** Ing-Marie Olsson, Charlotta Dykes, Ulrika Olsson-Möller, Marlene Malmström.

**Writing – review & editing:** Ing-Marie Olsson, Lisa Rydén, Ulrika Olsson-Möller, Marlene Malmström.

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
