## [Decision Letter · Decision Letter 0]

29 Oct 2024

PONE-D-24-21825Experiences of rehabilitation one year after breast cancer diagnosis - A focus group study from the ReScreen randomized controlled trialPLOS ONE

Dear Dr. Olsson,

Thank you for submitting your manuscript to PLOS ONE. After careful consideration, we feel that it has merit but does not fully meet PLOS ONE’s publication criteria as it currently stands. Therefore, we invite you to submit a revised version of the manuscript that addresses the points raised during the review process.

We look forward to receiving your revised manuscript.

Kind regards,

Eugenio Paci, MD

Academic Editor

PLOS ONE

Journal Requirements:

1. When submitting your revision, we need you to address these additional requirements. Please ensure that your manuscript meets PLOS ONE's style requirements, including those for file naming. The PLOS ONE style templates can be found at https://journals.plos.org/plosone/s/file?id=wjVg/PLOSOne_formatting_sample_main_body.pdf and https://journals.plos.org/plosone/s/file?id=ba62/PLOSOne_formatting_sample_title_authors_affiliations.pdf 2. Thank you for stating in your Funding Statement: "The ReScreen study (MM) was funded in part by external funding through governmental funding of clinical research within the Swedish National Health Service(ALF funding) (award number: 200-Projekt0190), FORTE (https://forte.se/en/) (award number: 2020-00105), the Swedish Breast Cancer Association (https://brostcancerforbundet.se), the Cancer and Allergy Foundation (https://cancerochallergifonden.se) (award number: 220) and The Swedish Cancer Society (https://www.cancerfonden.se/om-oss/about) (award number: 230656FE). The funding bodies had no role in any components, from study design, to submission for publication. For the purpose of Open Access, the author has applied a CC BY public copyright licence to any Author Accepted Manuscript (AAM) version arising from this submission." Please provide an amended statement that declares *all* the funding or sources of support (whether external or internal to your organization) received during this study, as detailed online in our guide for authors at http://journals.plos.org/plosone/s/submit-now.  Please also include the statement “There was no additional external funding received for this study.” in your updated Funding Statement. Please include your amended Funding Statement within your cover letter. We will change the online submission form on your behalf. 3. We note that you have indicated that there are restrictions to data sharing for this study. For studies involving human research participant data or other sensitive data, we encourage authors to share de-identified or anonymized data. However, when data cannot be publicly shared for ethical reasons, we allow authors to make their data sets available upon request. For information on unacceptable data access restrictions, please see http://journals.plos.org/plosone/s/data-availability#loc-unacceptable-data-access-restrictions.  Before we proceed with your manuscript, please address the following prompts: a) If there are ethical or legal restrictions on sharing a de-identified data set, please explain them in detail (e.g., data contain potentially identifying or sensitive patient information, data are owned by a third-party organization, etc.) and who has imposed them (e.g., a Research Ethics Committee or Institutional Review Board, etc.). Please also provide contact information for a data access committee, ethics committee, or other institutional body to which data requests may be sent. b) If there are no restrictions, please upload the minimal anonymized data set necessary to replicate your study findings to a stable, public repository and provide us with the relevant URLs, DOIs, or accession numbers. Please see http://www.bmj.com/content/340/bmj.c181.long for guidelines on how to de-identify and prepare clinical data for publication. For a list of recommended repositories, please see https://journals.plos.org/plosone/s/recommended-repositories. You also have the option of uploading the data as Supporting Information files, but we would recommend depositing data directly to a data repository if possible. Please update your Data Availability statement in the submission form accordingly. 4. Your ethics statement should only appear in the Methods section of your manuscript. If your ethics statement is written in any section besides the Methods, please move it to the Methods section and delete it from any other section. Please ensure that your ethics statement is included in your manuscript, as the ethics statement entered into the online submission form will not be published alongside your manuscript.

Additional Editor Comments:

This qualitative study evaluated in a very accurate and well carried on plan of qualitative analysis the experience with Breast Cancer of women already participating in a Randomized controlled trial (RESCREEN). The described relation between the experimental study and the qualitative study design is, in my opinion, very original and interesting. The analysis considered the two main categories and four sub-categories offering an informative interpretation of the findings. . The relevance of a proactive individual-based action is shown as a plus in the women perception of the health care. I appreciated the paper clear presentation. However, i suggest to consider the comments by the two reviewers preparing the second version of the paper. The non-participation rates, as suggested by a reviewer should be better discussed and possibly interpreted in relation to the distress fo the women. As the other reviewer suggested, in breast cancer care the issue of body images shouldt be more considered and experience reported.

Reviewers' comments:

Reviewer's Responses to Questions

**Comments to the Author**

1. Is the manuscript technically sound, and do the data support the conclusions?

Reviewer #1: Yes

Reviewer #2: Yes

2. Has the statistical analysis been performed appropriately and rigorously? 

Reviewer #1: I Don't Know

Reviewer #2: Yes

3. Have the authors made all data underlying the findings in their manuscript fully available?

Reviewer #1: Yes

Reviewer #2: Yes

4. Is the manuscript presented in an intelligible fashion and written in standard English?

Reviewer #1: Yes

Reviewer #2: Yes

5. Review Comments to the Author

Reviewer #1: The results of the focus groups describe and underline very important themes in the field of oncological rehabilitation.

The study contributes to bringing evidence on the importance of developing clinical practice towards a rehabilitation that is not only global and multidimensional, but above all guided and individualised.

It seems to me that the need to combine validated screening tools, such as the DT, with a proactive and orienting intervention, which in clinical practice requires a reference figure for patients, has been well underlined.

Half of the eligible patients involved in the RCT participate in the focus groups. What reflections can be made on this with respect to the topic of patients' needs?

I would better specify the exclusion criterion for patients with cognitive impairment, as it is a rather frequent problem in people undergoing oncology rehabilitation.

I would better describe the decision support tool referred to in line 135

For clarity, I would place the interview questions in the article rather than in separate material as supporting information

Reviewer #2: Dear authors,

thank you for your contribution. Here are my suggestions.

- Introduction: you just cited body image within your focus of interest. However, this topic has not been explored. Please, add further details about it, knowing its impact on Quality of Life in cancer patients. See: Sebri et al., 2023; Sebri; Durosini, Pravettoni, 2023. Secondo, please, provide information regarding previous psyhcological interventions. See: Matchim et al., 2011; Mifsud et al., 2021. Third, please structure your hypothesis and goals in a well-done way.

- Data analysis: please, provide more information about what type of analysis has been conducted in line with literature. Have you followed Braun and Clarke, for example? Literature has to be provided.

References

- Sebri, V., & Pravettoni, G. (2023). Tailored psychological interventions to manage body image: an Opinion study on breast Cancer survivors. International Journal of Environmental Research and Public Health, 20(4), 2991.

- Sebri, V., Durosini, I., & Pravettoni, G. (2023). How to address the body after breast cancer? A proposal for a psychological intervention focused on body compassion. Frontiers in Psychology, 13, 1085837.

- Matchim, Y.; Armer, J.M.; Stewart, B.R. Effects of mindfulness-based stress reduction (MBSR) on health among breast cancer survivors. Western J. Nurs. Res. 2011, 33, 996–1016.

- Mifsud, A.; Pehlivan, M.J.; Fam, P.; O’Grady, M.; van Steensel, A.; Elder, E.; Gilchrist, J.; Sherman, K.A. Feasibility and pilot study of a brief self-compassion intervention addressing body image distress in breast cancer survivors. Health Psychol. Behav. Med. 2021, 9, 498–526.

6. PLOS authors have the option to publish the peer review history of their article (what does this mean?). If published, this will include your full peer review and any attached files.

Reviewer #1: **Yes: **elisa grechi

Reviewer #2: No

---

## [Author Response · Author response to Decision Letter 0]

18 Nov 2024

Additional Editor Comments:

This qualitative study evaluated in a very accurate and well carried on plan of qualitative analysis the experience with Breast Cancer of women already participating in a Randomized controlled trial (RESCREEN). The described relation between the experimental study and the qualitative study design is, in my opinion, very original and interesting. The analysis considered the two main categories and four sub-categories offering an informative interpretation of the findings. The relevance of a proactive individual-based action is shown as a plus in the women perception of the health care. I appreciated the paper clear presentation. However, I suggest considering the comments by the two reviewers preparing the second version of the paper. The non-participation rates, as suggested by a reviewer should be better discussed and possibly interpreted in relation to the distress fo the women. As the other reviewer suggested, in breast cancer care the issue of body images should be more considered and experience reported.

Answer: We sincerely thank the editor for the positive valuation of our study. We have adjusted our manuscript and developed the important areas suggested. 

Reviewer 1

The results of the focus groups describe and underline very important themes in the field of oncological rehabilitation.

The study contributes to bringing evidence on the importance of developing clinical practice towards a rehabilitation that is not only global and multidimensional, but above all guided and individualised.

It seems to me that the need to combine validated screening tools, such as the DT, with a proactive and orienting intervention, which in clinical practice requires a reference figure for patients, has been well underlined.

Answer: We thank the reviewer for their positive appraisal of the manuscript.

Reviewer 1: Half of the eligible patients involved in the RCT participate in the focus groups. What reflections can be made on this with respect to the topic of patients' needs?

Answer: Thank you for pointing at this important matter. We have further developed the reasoning in the discussion section, see page 22-23, lines 508-513.

Reviewer 1: I would better specify the exclusion criterion for patients with cognitive impairment, as it is a rather frequent problem in people undergoing oncology rehabilitation.

Answer: Thank you. Those inclusion and exclusion criteria were for the ReScreen RCT, in which all the patients in the present study had participated. That inclusion took place before the start of treatment. Therefore, patients who experienced cognitive impairment as a result of treatment were included in the ReScreen RCT and their needs were addressed in the intervention. At the inclusion into the present focus group study, the only inclusion criteria were that the participants had fulfilled their participation in the RCT, and that at most two years have passed since the beginning of treatment. No questions related to the former (i.e. ReScreen RCT) criteria were asked about. We have now clarified this in the manuscript (page 8, lines 183-185).

Reviewer 1. I would better describe the decision support tool referred to in line 135.

Answer: Thank you for pointing this out, the decision support tool is essential for the ReScreen intervention which the women in the IG had received. We have now outlined the steps that the decision support tool entailed (page 7, lines 146-149) and clarified that more detailed information about the decision support tool can be found in the ReScreen RCT protocol (page 7, lines 154-155).

Reviewer 1: For clarity, I would place the interview questions in the article rather than in separate material as supporting information.

Answer: Thank you, we have now included a table containing the semi-structured interview guide in the manuscript (Table 2, page 10).

Reviewer 2

Dear authors,

thank you for your contribution.

Answer: We thank the reviewer for their comments on the manuscript.

Reviewer 2: Introduction: you just cited body image within your focus of interest. However, this topic has not been explored. Please, add further details about it, knowing its impact on Quality of Life in cancer patients. See: Sebri et al., 2023; Sebri; Durosini, Pravettoni, 2023. 

Answer: Thank you for the comment and the suggested references. One of the consequences of breast cancer treatment is indeed negative body image. In the Background, we expanded on the topic to illuminate its importance for the affected women with breast cancer (page 4, lines 77-81). Still, the focus of our study was on women’s experiences of rehabilitation and not on their experiences of consequences from diagnosis and treatment, nor specific consequences potentially affecting quality of life in this situation. Body image was not discussed during the interviews, and it was only briefly mentioned by participants. Therefore, within this study we cannot draw any conclusions about body image. This is now clarified in the discussion (page 22, lines 489-492).

Reviewer 2: Secondo, please, provide information regarding previous psyhcological interventions. See: Matchim et al., 2011; Mifsud et al., 2021. 

Answer: Thank you. This is a topic described briefly in page 4, lines 91-92. Within this study we describe the need for tailored support, and navigation in health care system. That of course include psychological interventions, and in the background, we give a few examples, as well as examples for physical interventions. We have now increased this section with examples of psychological interventions which have had promising results on wellbeing among women with BC (page 4, lines 92-94). 

Reviewer 2: Third, please structure your hypothesis and goals in a well-done way.

Answer: Thank you. In this qualitative study our aim was to explore the experience of rehabilitation after participating in an RCT. We aimed to generate insights and did not aim to test predefined hypotheses. The variation between the groups was a finding in this process of analysis, not a predefined hypothesis or goal, in line with the qualitative research design1.

Reviewer 2: Data analysis: please, provide more information about what type of analysis has been conducted in line with literature. Have you followed Braun and Clarke, for example? Literature has to be provided.

Answer: Thank you for pointing this out. This has been clarified (page 11, lines 221-222).

1Polit, D.F., & Beck, C.T. (2017). Essentials of Nursing Research – Appraising evidence for nursing practice (9th ed.). Wolters Kluwer.

---

## [Decision Letter · Decision Letter 1]

2 Dec 2024

Experiences of rehabilitation one year after breast cancer diagnosis - A focus group study from the ReScreen randomized controlled trial

PONE-D-24-21825R1

Dear Dr. Olsson,

We’re pleased to inform you that your manuscript has been judged scientifically suitable for publication and will be formally accepted for publication once it meets all outstanding technical requirements.

Kind regards,

Eugenio Paci, MD

Academic Editor

PLOS ONE

Additional Editor Comments (optional):

All comments adressed

Reviewers' comments:

Reviewer's Responses to Questions

**Comments to the Author**

Reviewer #2: All comments have been addressed

2. Is the manuscript technically sound, and do the data support the conclusions?

Reviewer #2: Yes

3. Has the statistical analysis been performed appropriately and rigorously? 

Reviewer #2: Yes

4. Have the authors made all data underlying the findings in their manuscript fully available?

Reviewer #2: Yes

5. Is the manuscript presented in an intelligible fashion and written in standard English?

Reviewer #2: Yes

6. Review Comments to the Author

Reviewer #2: Dear authors, thank you for your revisions and the time-dedicated to them. I have no other comments to add.

7. PLOS authors have the option to publish the peer review history of their article (what does this mean?). If published, this will include your full peer review and any attached files.

Reviewer #2: No

---

## [Editor Report · Acceptance letter]

15 Jan 2025

PONE-D-24-21825R1 

PLOS ONE

Dear Dr. Olsson, 

I'm pleased to inform you that your manuscript has been deemed suitable for publication in PLOS ONE. Congratulations! Your manuscript is now being handed over to our production team.

Kind regards, 

on behalf of

Dr. Eugenio Paci 

Academic Editor

PLOS ONE